# Tracking Multidrug Resistance in Gram-Negative Bacteria in Alexandria, Egypt (2020–2023): An Integrated Analysis of Patient Data and Diagnostic Tools

**DOI:** 10.3390/antibiotics13121185

**Published:** 2024-12-05

**Authors:** Sascha D. Braun, Shahinda Rezk, Christian Brandt, Martin Reinicke, Celia Diezel, Elke Müller, Katrin Frankenfeld, Domenique Krähmer, Stefan Monecke, Ralf Ehricht

**Affiliations:** 1Leibniz Institute of Photonic Technology, Leibniz Centre for Photonics in Infection Research (LPI), 07745 Jena, Germany; martin.reinicke@leibniz-ipht.de (M.R.); celia.diezel@leibniz-ipht.de (C.D.); elke.mueller@leibniz-ipht.de (E.M.); stefan.monecke@leibniz-ipht.de (S.M.);; 2InfectoGnostics Research Campus Jena, Center for Applied Research, 07743 Jena, Germany; christian.brandt@med.uni-jena.de; 3Department of Microbiology, Medical Research Institute, Alexandria University, Alexandria 21561, Egypt; shahinda.rezk@alexu.edu.eg; 4Institute of Virology, Medical Center, University of Freiburg, 79104 Freiburg, Germany; 5Institute of Infectious Diseases and Infection Control, Jena-University Hospital, Friedrich Schiller University, 07747 Jena, Germany; 6INTER-ARRAY part of fzmb GmbH, 99947 Bad Langensalza, Germany; 7Institute of Physical Chemistry, Friedrich Schiller University Jena, 07743 Jena, Germany

**Keywords:** carbapenemase, multidrug-resistant bacteria, microarray, nanopore sequencing, Microreact

## Abstract

Background: The rise in carbapenem-resistant *Enterobacteriaceae* (CRE) in Egypt, particularly in hospital settings, poses a significant public health challenge. This study aims to develop a combined epidemiological surveillance tool utilizing the Microreact online platform (version 269) and molecular microarray technology to track and analyze carbapenem-resistant *Escherichia coli* strains in Egypt. The objective is to integrate molecular diagnostics and real-time data visualization to better understand the spread and evolution of multidrug-resistant (MDR) bacteria. Methods: The study analyzed 43 *E. coli* isolates collected from Egyptian hospitals between 2020 and 2023. Nanopore sequencing and microarray analysis were used to identify carbapenemase genes and other resistance markers, whereas the VITEK2 system was employed for phenotypic antibiotic susceptibility testing. Microreact was used to visualize epidemiological data, mapping the geographic and temporal distribution of resistant strains. Results: We found that 72.09% of the isolates, predominantly from pediatric patients, carried the *bla*NDM-5 gene, while other carbapenemase genes, including *bla*OXA-48 and *bla*VIM, were also detected. The microarray method demonstrated 92.9% diagnostic sensitivity and 87.7% diagnostic specificity compared to whole-genome sequencing. Phenotypic resistance correlated strongly with next-generation sequencing (NGS) genotypic data, achieving 95.6% sensitivity and 95.2% specificity. Conclusions: This method establishes the utility of combining microarray technology, NGS and real-time data visualization for the surveillance of carbapenem-resistant *Enterobacteriaceae,* especially *E. coli*. The high concordance between genotypic and phenotypic data underscores the potential of DNA microarrays as a cost-effective alternative to whole-genome sequencing, especially in resource-limited settings. This integrated approach can enhance public health responses to MDR bacteria in Egypt.

## 1. Introduction

The global emergence of multidrug-resistant (MDR) bacteria presents a significant threat to public health, undermining the effectiveness of antibiotics and leading to increased morbidity and mortality rates. In Egypt and other parts of North Africa, the emergence and spread of MDR organisms, particularly among Gram-negative bacteria like *Escherichia coli*, have become a critical concern in both hospital and community settings [1,2,3]. Hospitals are encountering higher incidences of *E. coli* strains resistant to multiple classes of antibiotics, including extended-spectrum beta-lactams (ESBL), fluoroquinolones, aminoglycosides and carbapenems [4,5,6]. The prevalence of ESBL-producing and carbapenem-resistant *E. coli* strains is particularly alarming due to the limited treatment options available for such infections. This situation poses a serious challenge to healthcare providers, as these resistant strains are associated with higher rates of treatment failure, increased healthcare costs and a greater risk of complications.

Additionally, the use of antibiotics in agriculture, particularly as growth promoters or for disease prevention in animal husbandry, selects for resistant bacteria that can be transferred to humans through the food chain [7,8]. Furthermore, humans working in close contact with animals, such as farm workers, can also serve as carriers of these resistant bacteria [9]. International travel and migration also contribute to the spread of resistant bacteria, introducing new resistance mechanisms into different regions [10,11,12]. Conflicts and wars exacerbate this crisis, as they create large pools of multidrug-resistant infections that can spread widely. For example, in Ukraine, there are clear signals that antimicrobial resistance has worsened since the war, with increased resistance patterns observed among both soldiers and civilians [13]. Similarly, the protracted conflict against ISIS has led to large-scale population displacements, contributing to the spread of resistance genes across borders [14]. These factors collectively create an environment conducive to the emergence and dissemination of MDR bacteria.

The impact of MDR bacteria on healthcare in Egypt and North Africa is profound. Infections caused by MDR *E. coli* strains are difficult to treat due to resistance to multiple antibiotic classes, necessitating the use of last-resort antibiotics that may be less effective or have more severe side effects, e.g., colistin [15]. This leads to limited treatment options for patients, increased healthcare costs due to longer hospital stays and additional diagnostic testing, and a higher risk of morbidity and mortality. Delays in administering [14] an effective therapy can result in worsened patient outcomes, including complications and death. The burden on healthcare systems is significant, with the need for isolation procedures and additional infection control measures straining limited resources [16].

Addressing the challenge of MDR bacteria requires a multifaceted approach. Strengthening antimicrobial stewardship is crucial, involving the implementation of programs to optimize antibiotic prescribing practices, including guidelines for empiric therapy and de-escalation based on culture results [17,18]. Enhancing infection control measures is essential, with improvements needed in hand hygiene, sterilization procedures and environmental cleaning in healthcare settings to prevent the spread of resistant organisms. Establishing national and regional surveillance systems to track resistance patterns and to promptly identify outbreaks is necessary for effective monitoring and response [19].

Given this critical situation, understanding the molecular mechanisms underlying antibiotic resistance and the epidemiology of MDR bacteria is essential for developing effective strategies to combat their spread. In this context, we conducted a study analyzing 43 *E. coli* strains isolated from different human samples collected in different hospitals in Egypt. All isolates exhibited phenotypic resistance to carbapenems, which are considered antibiotics of last resort for severe bacterial infections. Additionally, these strains showed resistance to numerous other antibiotics, highlighting their multidrug-resistant nature.

To comprehensively characterize the genotype and phenotype of these MDR *E. coli* strains, we employed a combination of advanced analytical techniques. Nanopore sequencing was utilized to obtain detailed genomic information, allowing for the identification of all (also new allelic) resistance genes and genetic elements associated with antibiotic resistance. The VITEK2 automated system was used to determine the phenotypic antibiotic susceptibility profiles of the isolates.

By comparing the data obtained from microarray analysis and sequencing, we aimed to assess the concordance between these genotypic detection methods. Microarray technology, while significantly less expensive and faster than sequencing, may offer an effective alternative for epidemiological surveillance, particularly in resource-limited settings. Evaluating the concordance rate between these diagnostic tools is crucial for validating their effectiveness and reliability in clinical settings. Moreover, integration of phenotypic data from the VITEK2 analysis with genotypic data enhances the understanding of the relationship between genetic determinants of resistance and their expression in clinical isolates.

An essential aspect of our study is the epidemiological approach. Utilizing Microreact, an open-source platform for data visualization and sharing, we aim to map the distribution and relationships of these isolates over time and geography (https://microreact.org/project/s6LetF1LjhmKdmU4hr97Gu-amr-egypt-overview, updated on 21 October 2024). By visualizing genomic and epidemiological data in an interactive format, we are able to identify patterns of transmission, potential outbreak clusters and the emergence of new resistance mechanisms. This approach facilitates long-term observation and monitoring of multidrug-resistant bacteria in Egypt, providing valuable and freely available insights for public health interventions.

Our study seeks to enhance the understanding of the molecular mechanisms driving antibiotic resistance in *E. coli* strains in Egypt and to establish a foundation for ongoing surveillance of MDR bacteria. While our analysis focused on just 43 different strains, this should be seen as a starting point for the development of the epidemiological platform, Microreact. By incorporating DNA microarray analysis as a fast and cost-effective alternative to sequencing for genotypic detection, as well as robust typing methods like multi-locus sequence typing (MLST) [20,21], we can optimize the balance between resource use and data acquisition.

We are aiming to collaborate with partners in Egypt [22] and other countries (i.e., Greece [23], Pakistan [24], Algeria or Romania [25]) to analyze more strains over time, allowing Microreact to expand its capacity and utility progressively. This approach could be particularly beneficial in settings where sequencing resources are limited, enabling broader surveillance coverage. By combining advanced genomic analysis, phenotypic characterization and open data visualization, we aim to contribute to the development of targeted strategies to monitor, control and prevent the dissemination of these resistant strains. Ultimately, this research supports the global effort to address the pressing issue of antibiotic resistance, emphasizing the importance of integrated approaches that combine molecular diagnostics, epidemiology and data sharing for effective disease control and prevention.

## 2. Results

A total of 43 bacterial isolates were analyzed to determine their O:H type, multi-locus sequence type (MLST), as well as their resistance genotype and phenotype. No O:H types were identified in the non-*Escherichia coli* strains included in this study. The isolates exhibited considerable variability across these molecular types, indicating diverse strain circulation within the study region and time frame.

Among *E. coli* isolates, the O:H type distribution revealed significant heterogeneity (Figure 1a). In total, 12 distinct O:H types were identified. The most frequent combination was O102:H6, accounting for 28.95% of all isolates (n = 11). Other common O:H types included O9:H30 (13.16%, n = 5), Onovel32:H9 (13.16%, n = 5) and O54:H28 (13.16%, n = 5). O:H types O11:H25 and O109:H12 were found only once each, in the youngest patients, with an age of 9 months and 6 years, respectively. Patients from whom isolates with the most abundant combination, O102:H6, originated had a mean age of 10.90 years (n = 11), suggesting its prevalence in older children or younger adolescents. O9:H30, another common combination, was primarily found in young adults, with a mean age of 22.39 years (n = 5), while O54:H28 was prevalent in adults, with a mean age of 29.21 years (n = 5). Similarly, O8:H9 (n = 3) was more prevalent in younger patients, with a mean age of 9.35 years. Conversely, certain O:H types were more common among adults. Onovel32:H9 (n = 5) was associated with older patients, with a mean age of 33.13. O16:H5 (n = 3) also showed a higher prevalence in adults, with a mean age of 24.81 years.

MLST analysis revealed significant genetic diversity among the *E. coli* isolates, with ST-405 being the most common sequence type, representing 26.83% (n = 11) of isolates and forming a distinct clade within the phylogenetic tree (Figure 2). Known for its association with multidrug resistance, ST-405 underscores its clinical relevance and was more common in younger individuals, with patients ranging from 3 days to 43 years old (Figure 1b). ST-131, another globally recognized multidrug-resistant lineage, was frequently detected (9.76% of isolates) and formed a well-defined cluster, indicating close evolutionary relationships. Other notable sequence types included ST-156 and ST-361 (12.20% each), ST-167 and ST-648, with the latter emerging as a concerning lineage associated with antimicrobial resistance. Less frequent types, such as ST-69 and ST-410, were each detected only once but are known for their resistance to multiple antibiotics, highlighting their potential clinical impact. The outgroup isolates from *Klebsiella pneumoniae* and *E. hormaechei*, included for phylogenetic comparison, showed clear species-level divergence from the *E. coli* isolates. The MLST profiles revealed that specific types, such as ST-405 and ST-156, were more prevalent in younger patients, emphasizing the need for targeted surveillance of high-risk lineages in diverse age groups.

Carbapenemase genes were detected in all isolates (n = 43) (Figure 2). The most frequently identified carbapenemase gene was *bla*NDM-5, present in 72.09% of isolates (n = 31), with a mean patient age of 21.23 years (range: 10 days to 74 years). This carbapenemase gene was predominantly associated with younger patients, highlighting its prevalence in pediatric populations (Figure 1c). *bla*NDM-5/*bla*OXA-48 was the second most common combination, identified in 9.30% of isolates (n = 4), with a mean patient age of 46.90 years (range: 1–76 years), suggesting its association with older individuals. *bla*NDM-1 was detected in 6.98% of isolates (n = 3), with a mean patient age of 31.92 years. Other carbapenemase genes included *bla*VIM-1 (4.65%, n = 2, mean age: 32.01 years), *bla*OXA-244 (2.33%, n = 1, age: 43 years) and *bla*OXA-181 (2.33%, n = 1). Interestingly, *bla*OXA-181 and the combination *bla*NDM-1/*bla*OXA-244 were both found in newborn patients (aged 3 days and 26 days, respectively), indicating early life infection risks for these carbapenemase types.

The most frequently identified ESBL gene was *bla*CTX-M-15, which was found in 55.81% of the isolates (n = 24). The mean age of patients with *bla*CTX-M-15-positive isolates was 26.02 years (range: 3 days–76 years), indicating its prevalence across a broad age range. In addition, *bla*CTX-M-9 and its allelic variant *bla*CTX-M-27 were detected in 13.95% of the isolates (n = 6). These genes were associated with a mean patient age of 13.97 years, with most cases occurring in younger individuals. The age distribution for *bla*CTX-M-15 was broad, while *bla*CTX-M-9/CTX-M-27 combinations were more commonly observed in younger patients.

Narrow-spectrum beta-lactamase (NSBL) genes were also distributed across different age groups. *bla*OXA-1 was present in 13.95% of isolates (n = 6), with a mean patient age of 36.18 years. A combination of *bla*OXA-1 plus *bla*TEM-1 was found in younger patients, with a mean age of 8.62 years (n = 6), indicating its association with pediatric infections. *bla*TEM-1, the most frequently identified NSBL gene, was found in 25.58% of isolates (n = 11), with a mean patient age of 19.98 years. Notably, *bla*OXA-9; *bla*SHV-212; *bla*TEM-150 was found in an elderly patient (82 years), highlighting that the carriage of strains with certain NSBL combinations is linked to advanced age.

The performance of the CarbaResist genotyping kit was evaluated against the current molecular gold standard, full genome sequencing using nanopore technology across the 43 clinical isolates. The analysis yielded an overall sensitivity of 92.9%, specificity of 87.7% and diagnostic accuracy of 91.1%, demonstrating strong concordance between the microarray and sequencing results for the tested bacterial strains (Table 1; details of all 43 isolates are provided in Appendix A). However, variability in performance was observed in specific isolates, with *E. coli* **EG-98203**, *E. coli* **EG-98208** and *Enterobacter hormaechei* **EG-98213** showing the lowest accuracies.

Isolate **EG-98203** exhibited an accuracy of 80.0% (Figure 2), sensitivity of 81.3% and specificity of 75.0%. One of the key findings was a false negative result for the detection of the *bla*NDM-5 carbapenemase gene by the microarray, which was identified in the nanopore sequencing data. Additionally, *qnr*S1, a gene associated with quinolone resistance, was detected by nanopore sequencing but not by the microarray. However, the *mcr-*1 gene, associated with colistin resistance, was correctly identified by both the microarray and nanopore sequencing. Despite the agreement on *mcr*-1, the microarray over-detected resistance in some categories, including sulfonamide resistance (with false positives for *sul*1), leading to reduced specificity. The combination of missed detections of important resistance genes and over-detection of others contributed to the lower overall accuracy and specificity for this isolate.

Isolate **EG-98208** demonstrated an accuracy of 80.0%, with a sensitivity of 85.7% and specificity of 75.0%. The lower specificity was largely driven by false positives in the detection of aminoglycoside resistance genes, where the microarray detected genes not found in the nanopore sequencing results. For example, although the microarray detected *aadA2* accurately, it also identified additional aminoglycoside resistance genes not present in the sequencing data. False positives were also observed in genes related to the multidrug efflux pump and integrase genes, which were not identified in the nanopore sequencing. Despite these discrepancies, both methods correctly identified the *bla*NDM carbapenemase gene, highlighting the microarray’s reliability in this critical resistance category. However, the over-detection of other resistance genes led to significant reductions in both specificity and accuracy.

For isolate **EG-98213**, the accuracy was 80.0%, with sensitivity at 77.8% and specificity at 83.3%. In this case, false negatives played a significant role in lowering sensitivity, particularly in the detection of multidrug efflux pump genes, such as *oqxA*5 and *oqxB*9, which were identified by nanopore sequencing but missed by the microarray. Additionally, aminoglycoside resistance genes like *aph(3′)-VIb* were detected by nanopore sequencing but not by the DNA microarray. Conversely, a false positive for *mcr-9*, a colistin resistance gene, was found in the microarray results but not in the nanopore sequencing, reducing specificity. The combination of undetected resistance genes by the microarray and the false detection of the colistin resistance gene resulted in lower sensitivity and specificity for this isolate.

The comparison between phenotype (VITEK2) and genotype (nanopore sequencing) resistance profiles across various antibiotic classes revealed strong concordance, with high overall performance metrics: a sensitivity of 95.6%, specificity of 95.2% and accuracy of 95.4% (Table 1). These results reflect the ability of genetic analysis to predict phenotypic resistance across multiple antibiotic classes effectively, although there were a few notable discrepancies in specific cases (details of all 43 isolates are provided in Appendix A).

Carbapenem resistance, mediated by carbapenemase genes like *bla*NDM-5, *bla*OXA-48, *bla*VIM-1 and *bla*OXA-181, showed strong overall concordance between genotype and phenotype. In isolates such as **EG-98190** and **EG-98229**, which harbored *bla*NDM-5 or *bla*OXA-48, there was complete agreement between the detected presence of these carbapenemase genes and phenotypic resistance to imipenem and meropenem. Both isolates were resistant to the tested carbapenems, as expected from their genotypic profiles. Additionally, **EG-98195** and **EG-98231**, which carried both *bla*NDM-5 and *bla*OXA-48, also exhibited resistance to imipenem and meropenem in phenotypic testing, further reinforcing the robustness of genetic detection for carbapenem resistance. However, there was one notable exception, namely **EG-98192**, which carried the *bla*OXA-181 carbapenemase gene. Phenotypic testing showed susceptibility to both imipenem and meropenem, resulting in a false negative for carbapenem resistance. In contrast, **EG-98202,** which carried *bla*NDM-1 and *bla*OXA-244, exhibited resistance to both imipenem and meropenem, as expected, showing perfect concordance between genotype and phenotype. Similarly, **EG-98203**, harboring *bla*NDM-5, showed resistance across carbapenems, aligning well with its genotypic profile. Despite one exception, the overall concordance between genotype and phenotype in predicting carbapenem resistance remained strong across the majority of isolates tested.

For cephalosporins, the genotype correctly predicted resistance patterns across all isolates carrying extended-spectrum beta-lactamase (ESBL) alleles, such as *bla*CTX-M and *bla*TEM. These genes correlated strongly with resistance to cefuroxime, cefotaxime and ceftazidime. For example, **EG-98194**, which harbored *bla*CTX-M-15, *bla*CTX-M-27 and *bla*TEM-150, exhibited resistance to all tested cephalosporins, as expected from the genotype results. Similarly, **EG-98209**, carrying *bla*CTX-M-27, *bla*OXA-1 and *bla*TEM-1, showed resistance across multiple beta-lactams in both phenotype and genotype. This strong correlation indicates that the genetic detection of ESBL genes is highly reliable for the prediction of Ceph III resistance in *E. coli*.

Resistance to aminoglycosides, particularly gentamicin, was accurately predicted in isolates with aminoglycoside resistance genes, such as *aadA2*, *aac(6′)-Ib* and *aph(3″)-Ib*. For instance, isolates **EG-98197** and **EG-98209**, which harbored these genes, displayed resistance to gentamicin as expected, with no false positives or negatives. The presence of these genes correlated directly with phenotypic resistance, further confirming the high predictive value of the genotype for aminoglycoside resistance.

Trimethoprim/sulfamethoxazole resistance: The genotype also accurately predicted resistance to trimethoprim/sulfamethoxazole, with genes such as *dfrA12, sul1* and *sul2* being highly indicative of resistance. Isolates **EG-98210** and **EG-98192,** which carried these genes, showed consistent resistance to trimethoprim/sulfamethoxazole in both genotype and phenotype. No false positives or false negatives were observed, indicating strong concordance between the presence of these resistance genes and the phenotypic outcome.

Resistance to quinolones, specifically ciprofloxacin, was generally well predicted by the genotype. Resistance genes like *qnrS1* and *aac(6′)-Ib-cr* were associated with phenotypic resistance in most cases. For instance, **EG-98209**, which harbored *qnrS1,* displayed resistance to ciprofloxacin in both the genotype and phenotype. However, discrepancies were noted in **EG-98222**, which carried *aac(6′)-Ib-cr* but remained susceptible to ciprofloxacin in VITEK2 testing, resulting in a false positive for quinolone resistance. These occasional discrepancies suggest that while the presence of resistance genes correlates with phenotypic resistance, not all genes may consistently confer resistance in clinical testing. In the case of **EG-98231**, although this isolate exhibited phenotypic resistance to tigecycline, no known resistance genes specific to tigecycline were identified by either NGS or the microarray. Colistin resistance was correctly predicted by the presence of *mcr-1* in isolates like **EG-98203**, which exhibited resistance in both phenotype and genotype. However, a false positive was noted in **EG-98213**, where the genotype detected *mcr-9*, but the isolate remained susceptible to colistin in VITEK2 testing. While the overall accuracy of genotype prediction was high, a few specific isolates exhibited discrepancies between the phenotype and genotype. **EG-98214** and **EG-98216** both showed lower specificity (57.1%) due to false positives in sulfonamide resistance. These isolates carried *sul1* and *sul2*, which may not always translate to phenotypic resistance. **EG-98231** demonstrated two false positives in resistance to quinolones and fosfomycin. Despite carrying *qnrS1* and *fosA6*, the VITEK2 results indicated susceptibility, leading to discrepancies in these cases.

A phylogenetic tree was constructed to investigate evolutionary relationships among all 43 bacterial isolates (Figure 2). Multi-locus sequence typing (MLST) was used to define the sequence types (STs), with color-coded markers indicating the respective types. The sequence types identified include ST11, ST23, ST69, ST90, ST101, ST131, ST156, ST167, ST361, ST405, ST410, ST457, ST617, ST624 and ST648. The antimicrobial susceptibility profiles of the isolates were displayed in a heatmap (Figure 2), which demonstrated strong correlations between specific MLST types and resistance phenotypes. ST405 exhibited widespread resistance to multiple antibiotic classes, including third-generation cephalosporins, fluoroquinolones and carbapenems. This high level of resistance is consistent with its characterization as a multidrug-resistant lineage. ST131, while less frequent than ST405, showed a similarly broad resistance profile, reinforcing its role as a major global lineage of concern. ST167 and ST457 also demonstrated resistance to multiple antibiotics, aligning with their reported association with multidrug-resistant infections. Although ST69 and ST410 were detected only once, both isolates displayed resistance to several antibiotics, emphasizing the clinical relevance of these sequence types, even when present at low frequencies. In terms of infection origin, hospital-acquired isolates were predominantly associated with ST405 and ST131, reflecting the high resistance burden carried by these types. Community-acquired strains were more genetically diverse, encompassing ST648, ST457 and other types, suggesting that different sequence types contribute to infections across clinical settings.

## 3. Discussion

The results of our study provide a critical foundation for understanding the molecular mechanisms driving antibiotic resistance in *Escherichia coli* and other Gram-negative isolates in Egypt. By leveraging tools such as DNA microarray analysis and robust typing methods, including MLST, we demonstrated an efficient and scalable approach to characterizing bacterial clones and their population structure. Our findings reveal significant insights into the age distribution, molecular characteristics and carbapenemase gene profiles of the isolates, with the universal presence of carbapenemase genes underscoring the gravity of carbapenem resistance in the region. Specifically, *bla*NDM-5 emerged as the dominant carbapenemase gene, detected in 72.09% of isolates, aligning with regional trends reported in Egypt and North Africa [26,27].

The presence of *bla*NDM genes in combination with specific MLST types and their correlation with patient age provides crucial insights into the epidemiology of carbapenem-resistant *Escherichia coli*. The *bla*NDM (New Delhi Metallo-β-lactamase) family of genes is one of the most significant carbapenemase-producing genes globally, and its variants, such as *bla*NDM-1 and *bla*NDM-5, have been increasingly linked to multidrug-resistant (MDR) Enterobacterales in Egypt and other parts of North Africa [28,29]. In our study, the *bla*NDM-5 gene was found in 72.09% of isolates, and it was predominantly associated with younger patients (mean age: 21.23 years), including infants as young as 10 days old. This finding is consistent with reports that *bla*NDM-producing strains often affect younger populations, particularly in regions with high rates of healthcare-associated infections (HAIs). The pediatric vulnerability to these infections could be due to several factors, including underdeveloped immune systems and frequent hospitalizations (except for newborns), which increase exposure to resistant strains [30]. The high prevalence of *bla*NDM-5 in pediatric patients, especially those in neonatal intensive care units, reflects the growing concern over nosocomial transmission in hospital environments. In North Africa, studies have reported hospital outbreaks of NDM-producing *E. coli* and *Klebsiella pneumoniae*, with newborns and young children being particularly affected [27,30]. In our dataset, the youngest patients with *bla*NDM infections highlight the broad age range of those at risk, though younger patients are disproportionately affected. This underscores the need for enhanced infection prevention measures in healthcare settings to mitigate the spread of these highly resistant strains.

The MLST typing in our study revealed several known high-risk clones, such as ST-405 and ST-131, which were commonly associated with *bla*NDM-5. ST-405 was particularly frequent in younger patients, which is concerning, as it is a recognized high-risk clone associated with multidrug resistance. ST-405 has been identified in multiple studies as part of the global dissemination of resistance genes and has emerged as an important sequence type, particularly in Europe, where it has been linked to clinical infections and concerning antibiotic resistance patterns. Its presence in both young and adult populations in our study suggests both community and hospital transmission [31,32]. In Egypt and other parts of North Africa, ST-405 has been reported in multiple studies, often in association with *bla*NDM genes, further indicating that this clone is a significant driver of carbapenem resistance in the region. The ST-131 clone, another high-risk multidrug-resistant clone, was also detected in our study, though it was more commonly associated with older patients. ST-131 is recognized globally as a predominant lineage among extraintestinal pathogenic *E. coli* (ExPEC) isolates. It has been linked to the production of extended-spectrum β-lactamases (ESBLs), particularly CTX-M-15, and is often resistant to fluoroquinolones. This sequence type is primarily of serotype O25 and has been associated with significant outbreaks and infections, making it a critical public health concern. Its higher frequency in older patients in our study may suggest age-specific transmission dynamics, where younger populations are more susceptible to ST-405, while ST-131 predominates in adults, potentially due to differing exposure patterns or healthcare interactions.

The *bla*NDM-5 variant has gained attention due to its increased carbapenem hydrolyzing activity compared to *bla*NDM-1 [33]. First described in 2011 in a strain of *E. coli* from the UK [34], *bla*NDM-5 has been noted for its rapid global dissemination since then. It is more commonly detected in *E. coli*, particularly in specific sequence types like ST648, ST131 and ST405 [35], which were also found in our study. Its rapid spread in North Africa, including Egypt, is of concern due to the limited treatment options available for infections caused by NDM-producing Enterobacterales. The plasmid-mediated nature of *bla*NDM genes facilitates their dissemination across diverse settings by allowing easy transfer between different bacterial species and strains, complicating the epidemiology of infections and making control efforts increasingly difficult, particularly in resource-limited healthcare environments with inadequate infection control and antimicrobial stewardship. This is reflected in the spread of *bla*NDM-5 across age groups, as seen in our study, and the correlation of *bla*NDM genes with specific MLST types like ST-405 and ST-131 in younger patients, suggesting that these high-risk clones are established in both community and hospital settings. These trends align with broader patterns observed in Africa, where NDM-producing bacteria are frequently isolated from hospital outbreaks, especially in regions with weaker infection control practices and overburdened healthcare systems. Given the broad age range affected by *bla*NDM-positive isolates, especially in vulnerable populations, such as infants, targeted interventions are crucial. Neonatal and pediatric units are at a higher risk of nosocomial transmission, underscoring the need for proactive measures. To mitigate the spread of CRE infections in these settings, hospitals should enhance infection control through strict hand hygiene, device-related infection prevention, targeted antibiotic stewardship, proactive screening and isolation, as well as specialized training for healthcare workers and parent education on infection prevention [36,37].

The presence of *bla*OXA-48 in 9.30% of isolates further highlights the heterogeneity of carbapenemase resistance mechanisms. Studies from Egyptian hospitals have similarly reported OXA-48-type carbapenemases in *E. coli* and *K. pneumoniae* isolates, with particular concern over their rapid spread in healthcare-associated infections [38,39]. Interestingly, we identified both *bla*NDM-1 and *bla*OXA-181 in newborns, suggesting that these resistant strains are infiltrating even the most vulnerable populations in neonatal wards. This raises concerns about vertical transmission or hospital-acquired infections (HAIs), a pattern that has been observed in studies of neonatal sepsis across Africa [30].

Community and healthcare-associated transmission of carbapenem-resistant bacteria is a significant concern in Egypt and across North Africa. Carbapenem-resistant organisms are mostly associated with healthcare settings, where poor infection control practices can facilitate the spread of resistant strains, especially in intensive care units (ICUs) and neonatal units. However, increasing evidence shows that community-acquired infections are also becoming a prominent pathway for transmission. This suggests that carbapenemase-producing strains are not confined to hospitals but are being carried asymptomatically by individuals within the broader population, facilitating their circulation and spread beyond healthcare settings [40]. We identified five cases of community-acquired infections with carbapenemase-producing isolates, accounting for 11.63% of the total isolates in our study. In Egypt, studies indicate that healthcare-associated infections (HAIs) are the predominant source of carbapenem-resistant bacteria, accounting for over 80% of cases, particularly in ICUs and urology departments [26]. However, community-acquired infections are also on the rise, likely due to asymptomatic colonization in healthy individuals, which facilitates the silent spread of multidrug-resistant organisms (MDROs) between healthcare institutions and the community. This dual nature of transmission complicates efforts to control the spread of carbapenem-resistant bacteria because it extends the problem beyond the controlled environment of hospitals into the general population. In our study, the *bla*VIM-1 and *bla*NDM-5 variants were detected in both community and hospital settings. These genes are often carried on plasmids, making them highly transferable between different bacterial species, which accelerates their spread [41,42]. In our study, NDM-5-producing *E. coli* (**EG-98203**) were found to harbor additional resistance genes, such as those conferring resistance to colistin, further limiting treatment options [43]. To combat the spread of NDM-producing strains, enhanced infection control measures are critical. This includes rigorous screening protocols for at-risk populations (such as ICU and neonatal patients), antimicrobial stewardship programs and improved hygiene practices in both healthcare and community settings.

The MLST results from this study highlight critical insights into the genetic diversity of *E. coli* isolates and their connection to both age and antibiotic resistance patterns. The presence of high-risk sequence types (STs), such as ST-405 and ST-131, is particularly significant, as these are linked to varying resistance profiles and age-specific infection risks. ST-405, which was the most frequent sequence type, was predominantly found in younger patients, echoing global observations, where this type is associated with MDR strains and HAIs in neonatal and pediatric units [44,45]. This is consistent with reports from Europe (Greece, Italy), where ST-405 is commonly linked to hospital outbreaks of MDR *E. coli*, often carrying genes like *bla*NDM [46,47].

On the other hand, ST-131, another globally recognized high-risk clone, was more prevalent among adult patients, known for causing community-acquired infections, such as urinary tract and bloodstream infections. ST-131 is frequently associated with extended-spectrum beta-lactamases (ESBLs) and carbapenemase-producing genes, including *bla*NDM, making it highly resistant to a broad spectrum of antibiotics [48]. The age-related distribution of these types suggests that younger patients are more susceptible to ST-405, potentially due to hospital exposures, while ST-131 affects older adults, likely from both community and healthcare settings [49].

The co-occurrence of these high-risk clones with carbapenemase genes like *bla*NDM-5 in younger patients underlines the critical need for early detection and constant surveillance, especially in neonatal and pediatric wards, to prevent the spread of these resistant strains. These data emphasize the importance of targeted surveillance and infection control measures, particularly in healthcare environments where these resistant strains circulate. Overall, the integration of MLST data into understanding antibiotic resistance patterns across age groups provides valuable insights for developing tailored approaches to managing and controlling resistant infections.

The microarray-based detection system employed in this study demonstrated high levels of specificity and sensitivity, making it a valuable tool for rapid identification of antimicrobial resistance (AMR) genes. Compared to whole-genome sequencing (WGS), the microarray-based system offers advantages in terms of turnaround time and cost effectiveness while maintaining a robust performance in detecting critical resistance mechanisms. The dataset analyzed here supports these findings, with sensitivity recorded at 92.9% and specificity at 87.7%, consistent with results from similar studies on microarray applications in clinical microbiology [22,23,25].

A notable discrepancy was observed in the detection of the *mcr-9* gene when comparing the WGS and microarray results. The microarray detected the presence of *mcr-9*, while WGS failed to identify the complete gene, detecting only a small part of it on the chromosome. This difference can be attributed to several key factors, which highlight the complementary strengths and limitations of these two detection methods. In the microarray analysis, two probe–primer pairs were used for detection: one perfectly matched this region, while the other did not. Based on these results, we recommend that a positive signal for *mcr-9* should only be considered valid if both probe–primer pairs generate a signal, not just one. This would explain why the microarray detected *mcr-9* despite the partial identification by WGS. Moreover, the partial detection of the gene also explains the absence of the phenotypic resistance, as the incomplete gene may lack the functional regions necessary for conferring resistance.

One key advantage of the microarray approach is its speed. While WGS offers comprehensive insight into bacterial genomes, including novel or rare resistance genes, it requires more time and computational resources. Microarrays, in contrast, allow for the simultaneous screening of multiple genes within one working day, providing timely results that are crucial for infection control and treatment decisions in clinical settings. This rapid detection capability is underscored by Braun et al. (2014), who reported that the microarray method could detect carbapenemase genes such as *bla*VIM and *bla*NDM with over 98% accuracy within a short time frame [50]. Similarly, the microarray system’s ability to handle high-throughput testing makes it ideal for clinical and surveillance environments, where speed and efficiency are paramount [51].

Moreover, microarrays provide a practical and affordable alternative for large-scale screening. Although sequencing remains undoubtedly the gold standard for genomic analysis, its cost and resource demands can be prohibitive, especially for routine diagnostics or in resource-limited settings. Microarrays, on the other hand, offer a cost-effective solution, as they can screen for multiple resistance markers without the need for extensive bioinformatics expertise. Studies like those conducted by Gwida et al. (2020) and Geue et al. (2014) further confirm the versatility of microarrays in detecting a broad range of AMR and virulence genes [22,52]. However, one limitation of microarrays is their reliance on pre-defined probes, which may result in the omission of novel or unexpected resistance genes. In contrast, sequencing can provide a more comprehensive view of the genome, detecting mutations that microarrays may overlook. Nonetheless, microarray technologies have evolved to include probes for the most clinically relevant genes, significantly reducing this limitation. In this study, the microarray accurately identified all major resistance genes, supporting its utility as a rapid diagnostic tool.

The comparison between phenotype and genotype resistance profiles has become a critical area in antimicrobial resistance research, especially with the growing reliance on WGS for predicting phenotypic resistance patterns. Our study demonstrated strong concordance between the two methods, with 95.6% sensitivity, 95.2% specificity and 95.4% accuracy. These metrics reflect the high predictive power of genotypic analysis in detecting phenotypic resistance across various antibiotic classes. The strong concordance we observed is consistent with other studies on genotype–phenotype correlations. For instance, a study on *Enterobacterales* isolates reported high concordance between genotype and phenotype, particularly for carbapenems, where WGS predicted resistance with 96% sensitivity and 97% specificity [53,54]. These findings suggest that genotypic methods, especially when focusing on well-characterized resistance genes like carbapenemases, offer a robust tool for predicting resistance patterns, albeit with certain limitations. Similarly, a study on *Salmonella enterica* isolates found that genotypic resistance predictions for beta-lactams, aminoglycosides and phenicol antibiotics were 96.2% concordant with phenotypic results [55]. This aligns well with our findings for aminoglycoside and beta-lactam resistance, where genetic markers like *aadA2*, *bla*CTX-M and *bla*TEM accurately predicted phenotypic outcomes.

Despite the high overall concordance, our study identified some discordances in carbapenem resistance prediction. Notably, isolate **EG-98192**, which carried *bla*OXA-181, showed phenotypic carbapenem susceptibility. The *bla*OXA-181 gene is part of the OXA-48-like family of carbapenemase genes, typically conferring resistance by hydrolyzing carbapenems [56]. However, the resistance patterns can vary depending on factors such as the expression level of *bla*OXA-181, which might be low if plasmid-related, leading to insufficient carbapenemase production [56,57]. In our study, OXA-181 was located on a large, possibly low-copy, plasmid of approximately 90 kb in size, which supports this hypothesis. Resistance is often amplified when *bla*OXA-181 is paired with porin loss, which restricts carbapenem entry into the bacterial cell. Without porin loss, *bla*OXA-181 alone may not result in manifest resistance. The discrepancy in EG-98192 could stem from low *bla*OXA-181 expression or the lack of synergistic resistance mechanisms, such as porin mutations [58]. This case illustrates the complexity of correlating genotype and phenotype, especially with blaOXA-181. This highlights the importance of considering broader genetic and cellular contexts when interpreting resistance predictions based on genetic data alone.

For cephalosporins and aminoglycosides, our study showed near perfect concordance between genotype and phenotype. Similar results have been widely reported, with studies consistently showing that the presence of genes such as *bla*CTX-M and *aadA2* accurately predicts phenotypic resistance [23,25,50]. This reinforces the reliability of genotypic testing in these classes of antibiotics, where specific resistance genes are well understood and lead to consistent phenotypic outcomes.

Quinolones, particularly ciprofloxacin, presented some challenges in our study, with occasional false positives, such as in isolate **EG-98222**, which carried *aac(6′)-Ib-cr* but remained susceptible to ciprofloxacin in phenotypic testing. This issue has been noted in other research works as well, where resistance genes like qnrS1 did not consistently confer phenotypic resistance unless combined with additional mutations or resistance mechanisms [59]. While genotypic predictions show high accuracy, it is important to note that resistance mechanisms are often multifactorial. Factors like gene expression, regulatory elements and the presence of compensatory mutations can influence whether a genotype translates into a resistant phenotype.

Building on the findings discussed above, which highlight the widespread presence of carbapenemase genes, the age-specific distribution of high-risk clones and the critical role of early detection in controlling multidrug-resistant organisms, it becomes clear that an effective combination of different diagnostic and epidemiological tools is essential in combating these threats. While whole-genome sequencing (WGS) offers comprehensive insights into resistance mechanisms, the resource-intensive nature of this method limits its accessibility in many regions. This is where alternative diagnostic approaches, such as microarray-based systems, demonstrate their value, especially in regions where sequencing technologies may not be readily available or affordable.

The microarray-based detection system utilized in this study demonstrates its value as a rapid, cost-effective and reliable tool for identifying antimicrobial resistance (AMR) genes. Its strong correlation with whole-genome sequencing (WGS) and phenotypic data underscores its utility for epidemiological surveillance, resistance pattern analysis and clinical decision making. Furthermore, the application feasibility of these developed approaches extends beyond carbapenem resistance to monitoring resistance against other antimicrobial drugs. This adaptability makes them highly relevant for broader AMR surveillance, addressing multidrug-resistant pathogens across diverse healthcare settings.

In resource-limited settings, where sequencing technologies may be inaccessible, this microarray approach offers a practical and affordable alternative, especially for monitoring high-risk patients and managing multidrug-resistant infections. Complementing this, the integration of molecular diagnostics with epidemiological tools like Microreact enables real-time data visualization and fosters collaborative efforts across borders. Although this study’s application of these tools—using data from 43 strains—represents only a starting point (proof of concept), it highlights the potential for expanding this scalable framework to enhance surveillance, guide interventions and contribute to the global fight against antimicrobial resistance. Future partnerships and broader implementations will further refine and strengthen this approach, particularly in regions where AMR poses a critical threat to public health.

## 4. Materials and Methods

The bacteria used in this study included *Escherichia coli*, *Klebsiella pneumoniae* and *Enterobacter hormaechei*. The bacterial isolates were collected from two hospitals in Alexandria, Egypt, between the years 2020 and 2023. Isolates numbered 1–24 were obtained from pediatric patients (ages ranging from 3 days to 6 years), while isolates numbered 25–43 were collected from adult patients (ages ranging from 19 to 82 years). Samples were taken from various sources, including urine, blood, swabs and aspirates, with each isolate identified by species, source of infection and patient demographics. The isolates were then categorized based on their healthcare-associated (HA) or community-acquired (CA) status, as well as the geographic distribution of the patients (urban or rural) (Table 2).

Bacterial isolates were cultured overnight at 37 °C on Columbia blood agar, and genomic DNA was extracted using the MACHEREY-NAGEL NucleoSpin Microbial DNA kit (MACHEREY-NAGEL, Düren, Deutschland) following the manufacturer’s instructions. After extraction, the DNA was prepared for sequencing using the 1D Genomic DNA Ligation Kit (SQK-NBD114.24; Oxford Nanopore Technologies, Oxford, UK) according to the manufacturer’s protocol. Before library preparation, DNA samples were subjected to size selection using AMPure beads (Beckman Coulter, Krefeld, Germany) at a 1:1 (*v*/*v*) ratio. Approximately 600 ng of DNA per sample, quantified using a Qubit 4 Fluorometer (Thermo Fisher Scientific, Waltham, MA, USA), was used for library preparation before being loaded onto an Oxford Nanopore Technologies flow cell (FLO-MIN114, containing R10.4.1 pores). Sequencing was conducted on a MinION Mk1B device for 90 h, starting with a minimum of 900 active pores, and managed through MinKNOW software (v24.02.16). Basecalling was performed using the dorado basecaller (v0.7.3) with the model res_dna_r10.4.1_e8.2_400bps_sup@2023-09-22_bacterial-methylation, producing FastQ files with 4000 reads per file. For barcode trimming, dorado demux (v0.7.3) with the same model was used. Contig assembly of quality-filtered sequence reads for each strain was conducted using flye software (v2.9.1). The assembled contigs underwent polishing in two steps: first, using racon (v1.5.0) iteratively for four rounds with the parameters match 8, mismatch 6, gap 8 and window lengths of 500, followed by medaka (v1.7.0) for the final polishing, using the model r1041_e82_400bps_sup_v4.3.0. The resulting corrected assemblies were used for further analysis (all FASTA files are provided in Appendix A).

MLST typing was performed using the mlst tool (v2.23.0) [60], and resistance gene analysis was conducted using abricate (v1.0.0), both developed by Torsten Seemann (https://github.com/tseemann, accessed on 5 October 2024). The databases used for resistance gene identification included the NCBI AMRFinderPlus database [61], containing 7010 genes (last updated on 27 September 2024), and the ResFinder database [62], consisting of 3194 genes (last updated on 27 September 2024). These databases ensured comprehensive and up-to-date detection of antimicrobial resistance determinants.

Whole-genome sequences were analyzed for single-nucleotide polymorphisms (SNPs) and subsequent phylogenetic tree reconstruction. Genomes were first clustered based on k-mer similarity using Sourmash (v4.8.11). A representative genome, 98204_Escherichia_coli, was randomly selected from the largest cluster and subsequently used as the reference strain for pairwise SNP identification with Snippy (v4.4.1; github.com/tseemann/snippy). A core genome SNP alignment was generated using Snippy-core (v4.4.1), focusing on regions conserved among all strains. The phylogenetic tree was reconstructed from the core SNP alignment using FastTree (v2.1.10), applying the generalized time-reversible (GTR) model for nucleotide substitution [63]. Visualization of the phylogenetic tree was performed via Microreact [64], which can be accessed via https://microreact.org/project/s6LetF1LjhmKdmU4hr97Gu-amr-egypt-overview, updated on 21 October 2024 for interactive exploration.

In parallel, genotyping of bacterial isolates was performed using the genotyping kit CarbaResist according to the manufacturer’s instructions (INTER-ARRAY part of fzmb GmbH, Bad Langensalza, Germany). The microarray platform enables DNA-based detection of the most common beta-lactamase and other resistance genes found in multidrug-resistant Gram-negative bacteria.

The already isolated RNA-free, low-fragmented genomic DNA, which was also used for whole-genome sequencing, was amplified and internally labeled with biotin-dUDP through a linear PCR amplification protocol utilizing antisense primers targeting specific resistance genes. This produced biotin-labeled single-stranded DNA (ssDNA), which was subsequently hybridized to an oligonucleotide microarray containing 230 different probes for carbapenemase genes, extended-spectrum beta-lactamases (ESBLs), AmpC beta-lactamases (AmpC) and other relevant antibiotic resistance genes. After hybridization, the microarray was washed to remove any non-specifically bound ssDNA, and hybridized biotin-labeled ssDNA was detected using HRP-conjugated streptavidin, followed by a colorimetric enzymatic reaction. The hybridization results were visualized, and spot intensities were automatically evaluated using the INTER-ARRAY Reader (INTER-ARRAY part of fzmb GmbH, Bad Langensalza, Germany). A digital image of the microarray was analyzed, and the data were compared against a reference database to determine the presence or absence of specific resistance genes, along with providing insights into the associated bacterial species.

To compare the results of sequencing with those obtained from the microarray, a gene-by-gene analysis was performed. The genes of interest were categorized into various resistance gene classes, such as carbapenemase genes, extended-spectrum beta-lactamase genes, other beta-lactamase genes, AmpC genes, genes associated with aminoglycoside, macrolide, quinolone, sulfonamide and trimethoprim resistance, as well as genes encoding virulence factors, multidrug efflux pumps, toxin–antitoxin systems and colistin resistance genes.

The detection results for each gene were classified into four categories: True Positives (TPs), where a gene was detected by both sequencing and microarray analysis; True Negatives (TNs), where no gene was detected by either method; False Positives (FPs), where a gene was detected by sequencing but not by the microarray; and False Negatives (FNs), where a gene was detected by the microarray but not by sequencing. These classifications were applied to each of the resistance gene categories listed.

For each category, the performance metrics were calculated based on TP, TN, FP and FN values. Sensitivity was calculated as the proportion of true positives relative to the total number of positives (TP + FN), while specificity was calculated as the proportion of true negatives relative to the total number of negatives (TN + FP). Concordance (or accuracy) was defined as the proportion of correctly classified genes (TP and TN) relative to the total number of classifications (TP + TN + FP + FN). These calculations were performed to assess the overall concordance between sequencing and microarray analysis in detecting antimicrobial resistance genes.

The bacterial phenotypes were determined using the VITEK2 system (bioMérieux, Nürtingen, Germany). Antimicrobial susceptibility testing was performed using the AST-N430 and AST-XN24 cards, following the manufacturer’s guidelines. Specifically, the AST-N430 card was used for testing a broad range of antibiotics, including colistin, while the AST-XN24 card was used for colistin-specific susceptibility. The inoculum was prepared by suspending colonies in 0.45% sodium chloride solution to reach a McFarland standard of 0.5. Then, the bacterial suspension was loaded into the VITEK2 system for automated analysis. The results were interpreted according to the latest EUCAST/CLSI breakpoints to classify isolates as susceptible, intermediate or resistant to colistin.

To compare phenotypic antimicrobial susceptibility results with genotypic resistance detection, a gene-to-susceptibility analysis was performed. The genes of interest were similarly grouped into various antimicrobial resistance categories as described above.

The detection results were classified into four categories: True Positives (TPs), where a resistance gene was detected genotypically, and phenotypic resistance was observed; True Negatives (TNs), where no resistance gene was detected, and the corresponding phenotype was susceptible; False Positives (FPs), where a resistance gene was detected genotypically, but phenotypic susceptibility was observed; and False Negatives (FNs), where no resistance gene was detected, but the phenotype displayed resistance. Sensitivity, specificity and accuracy were calculated as described above.

## 5. Conclusions

This study highlights the growing challenge of carbapenem-resistant *Escherichia coli* and emphasizes the importance of early detection, particularly in vulnerable populations, such as neonates and young patients. Both molecular methods—whole-genome sequencing (WGS) and DNA-microarray-based detection systems—demonstrated high sensitivity, specificity and accuracy in identifying antimicrobial resistance (AMR) genes. Specifically, the microarray method achieved 92.9% sensitivity and 87.7% specificity, closely aligning with WGS, while genotype–phenotype concordance exhibited 95.6% sensitivity, 95.2% specificity and 95.4% accuracy. These findings underscore the effectiveness of genotypic analysis in predicting antibiotic resistance patterns.

Our study aims to enhance the understanding of the molecular mechanisms driving antibiotic resistance in *E. coli* strains in Egypt and to establish a foundation for ongoing surveillance of multidrug-resistant (MDR) bacteria. While our analysis focused on just 43 different strains, this represents a starting point for developing the epidemiological platform Microreact. By incorporating DNA microarray analysis as a rapid and cost-effective alternative to sequencing for genotypic detection, alongside robust typing methods like multi-locus sequence typing (MLST), we can optimize the balance between resource use and data acquisition.

The microarray method proved to be an excellent alternative for detecting resistance genes, particularly in countries where WGS is not readily accessible due to high costs, long turnaround times or a lack of technical expertise. Its rapid detection capability, cost effectiveness and robust performance make it ideal for resource-limited settings, enabling timely clinical decisions and improving infection control strategies. Furthermore, we aim to collaborate with partners in Egypt and other countries, including Greece, Pakistan, Algeria and Romania, to analyze more strains over time, progressively expanding Microreact’s capacity and utility. This approach could be particularly beneficial in settings with limited sequencing resources, enabling broader surveillance coverage.

By integrating advanced genomic analysis, phenotypic characterization and open data visualization, we contribute to the development of targeted strategies for monitoring, controlling and preventing the dissemination of resistant strains. Ultimately, this research supports the global efforts to combat antibiotic resistance, emphasizing the importance of integrated approaches that combine molecular diagnostics, epidemiology and data sharing for effective disease control and prevention.

## Figures and Tables

**Figure 1 antibiotics-13-01185-f001:**
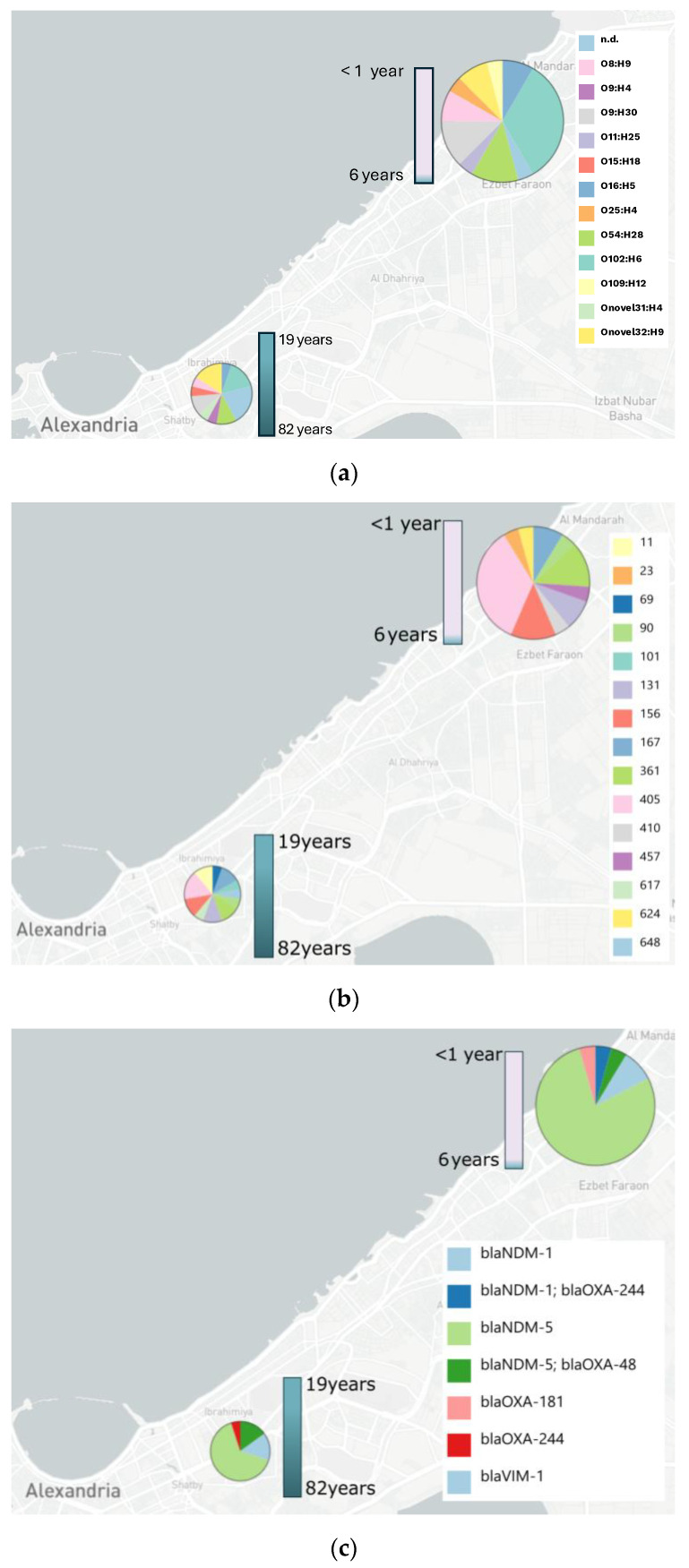
Geographic distribution and genotypic characterization of all 43 bacterial strains isolated from a pediatric ward (n = 24) and a standard ward or ICU from a city hospital (n = 19) in Alexandria. (**a**) O:H type distribution of all bacterial isolates (n.d.—not detected). (**b**) MLST distribution across two locations and the age of patients: pediatric station (<1 year and 6 years old) vs. adult patients (19 and 82 years old). (**c**) Carbapenemase gene distribution, showing the prevalence of different resistance genes (e.g., *bla*NDM, *bla*VIM) in bacterial isolates from pediatric and elderly patients in the respective wards.

**Figure 2 antibiotics-13-01185-f002:**
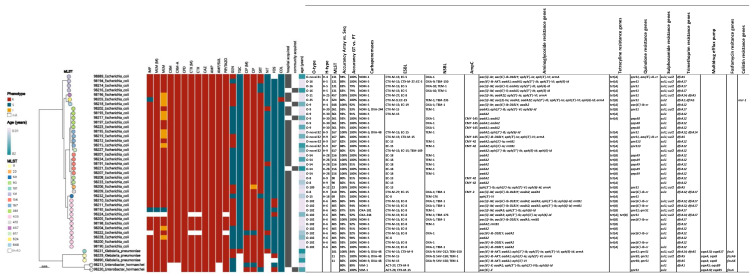
SNP-based phylogenetic clustering, MLST pattern, antimicrobial resistance profiles and patient age distribution of clinical isolates from pediatric and normal wards for elderly people. The dendrogram on the left shows phylogenetic relationships of *Escherichia coli*, *Klebsiella pneumoniae* and *Enterobacter hormaechei* isolates based on a pairwise SNP analysis. Multi-locus sequence typing (MLST) patterns are indicated directly on the tree next to each isolate (n.d.—not detected). The antimicrobial resistance profiles to various antibiotics are displayed in adjacent columns, with colors indicating resistance (red), susceptibility (blue), intermediate (yellow) and not determined (white). A gradient scale represents the patient age at the time of isolation (ranging from 0.01 to 82 years). Additionally, the table lists information on O-typing, H-typing, MLST, accuracy comparisons [microarray vs. whole-genome sequencing (WGS), genotype (GT) vs. phenotype (PT)] and detected resistance genes. For a more detailed view, please visit https://microreact.org/project/s6LetF1LjhmKdmU4hr97Gu-amr-egypt-overview, updated on 21 October 2024.

**Table 1 antibiotics-13-01185-t001:** Diagnostic specificity and sensitivity of the CarbaResist microarray in comparison with the full genome sequencing and the specificity and sensitivity of the genotype to phenotype measured by VITEK2.

Comparison	TP ^1^	FP	TN	FN	Specificity	Sensitivity	Accuracy
Microarray vs. Sequencing	421	29	207	32	87.7%	92.9%	91.1%
Genotype vs. Phenotype	346	9	177	16	95.2%	95.6%	95.4%

^1^ TP: True Positive, FP: False Positive, TN: True Negative, FN: False Negative.

**Table 2 antibiotics-13-01185-t002:** Demographic and clinical characteristics of bacterial isolates from pediatric and adult patients in two hospitals in Alexandria, Egypt.

Number	Strain-ID	Species	Source	Sex	Age	HA/CA ^1^	Geo. Distribution
1	**EG-98190**	*Escherichia coli*	swab	m	2 years	HA	rural
2	**EG-98191**	*Escherichia coli*	urine	m	3 months	HA	rural
3	**EG-98192**	*Escherichia coli*	blood	f	3 days	HA	rural
4	**EG-98193**	*Escherichia coli*	aspirate	m	9 months	HA	rural
5	**EG-98194**	*Escherichia coli*	urine	f	10 days	HA	urban
6	**EG-98195**	*Escherichia coli*	aspirate	f	7 months	HA	rural
7	**EG-98196**	*Escherichia coli*	aspirate	m	9 months	HA	rural
8	**EG-98197**	*Escherichia coli*	urine	m	40 days	HA	rural
9	**EG-98198**	*Escherichia coli*	urine	f	36 days	HA	rural
10	**EG-98200**	*Escherichia coli*	aspirate	f	2 years	HA	rural
11	**EG-98201**	*Escherichia coli*	urine	f	20 days	HA	rural
12	**EG-98202**	*Escherichia coli*	blood	m	26 days	HA	urban
13	**EG-98203**	*Escherichia coli*	aspirate	f	1 year	HA	urban
14	**EG-98204**	*Escherichia coli*	aspirate	m	5 years	HA	urban
15	**EG-98205**	*Escherichia coli*	urine	m	8 months	HA	rural
16	**EG-98206**	*Escherichia coli*	pus	m	6 years	HA	rural
17	**EG-98207**	*Escherichia coli*	swab	f	9 months	HA	rural
18	**EG-98208**	*Escherichia coli*	aspirate	m	3 years	HA	rural
19	**EG-98209**	*Escherichia coli*	swab	m	23 days	HA	rural
20	**EG-98210**	*Escherichia coli*	blood	f	30 days	HA	rural
21	**EG-98211**	*Escherichia coli*	urine	f	32 days	HA	rural
22	**EG-98212**	*Escherichia coli*	urine	m	25 days	HA	rural
23	**EG-98213**	*Enterobacter hormaechei*	swab	m	8 days	HA	urban
24	**EG-98214**	*Escherichia coli*	swab	m	5 months	HA	rural
25	**EG-98215**	*Escherichia coli*	urine	f	73 years	HA	urban
26	**EG-98216**	*Escherichia coli*	urine	m	74 years	CA	urban
27	**EG-98217**	*Escherichia coli*	urine	m	54 years	CA	rural
28	**EG-98218**	*Escherichia coli*	urine	m	49 years	CA	rural
29	**EG-98219**	*Escherichia coli*	urine	f	50 years	HA	rural
30	**EG-98221**	*Escherichia coli*	urine	f	81 years	CA	rural
31	**EG-98222**	*Escherichia coli*	urine	m	28 years	HA	rural
32	**EG-98223**	*Escherichia coli*	urine	m	57 years	HA	urban
33	**EG-98224**	*Escherichia coli*	urine	m	43 years	HA	rural
34	**EG-98227**	*Escherichia coli*	urine	f	42 years	HA	rural
35	**EG-98228**	*Escherichia coli*	urine	f	39 years	HA	rural
36	**EG-98229**	*Klebsiella pneumoniae*	swab	m	29 years	HA	rural
37	**EG-98230**	*Enterobacter hormaechei*	urine	m	64 years	CA	rural
38	**EG-98231**	*Klebsiella pneumoniae*	urine	f	82 years	HA	urban
39	**EG-98232**	*Escherichia coli*	urine	f	19 years	HA	urban
40	**EG-98233**	*Escherichia coli*	urine	f	25 years	HA	rural
41	**EG-98234**	*Escherichia coli*	urine	m	64 years	HA	rural
42	**EG-98889**	*Escherichia coli*	urine	f	76 years	HA	rural
43	**EG-98890**	*Klebsiella pneumoniae*	urine	f	76 years	HA	rural

^1.^HA—hospital-acquired; CA—community-acquired.

## Data Availability

Whole-genome sequences of all 43 bacterial strains are available on NCBI (https://www.ncbi.nlm.nih.gov/) using accession number PRJNA1180919; these also include the annotations for chromosomes and plasmids.

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
