# Peer review of "Tracking Multidrug Resistance in Gram-Negative Bacteria in Alexandria, Egypt (2020–2023): An Integrated Analysis of Patient Data and Diagnostic Tools"

_antibiotics, 2024, doi:10.3390/antibiotics13121185_

Round 1
Reviewer 1 Report
Comments and Suggestions for Authors
The work demonstrates scientific importance, being well designed and supported. However, the text, in some sections, presents redundancy. Here, I suggest some minor adjustments that should help.
1- The title suggests that the work aims to establish an Epidemiological Platform for identifying Carbapenem Resistance in Egypt using Microreact and Microarray for Real-Time Surveillance. But although the text emphasizes carbapenems, other classes are discussed and the importance of patient data, such as age, is evaluated. In fact, the work indicates, in a very relevant way, the epidemiological analysis of E. coli infection in Alexandria, Egypt, between 2020 and 2023 using phenotypic and genotypic tools. I suggest adapting the title.
2- A graph or table would be clearer to indicate the O/H distribution of E. coli.
3- Figure 1b is cited in the text as Figure 2 (Line 161).
4- To make the text easier, I suggest listing the cases of agreement and then the exceptions for each class of antimicrobials in relation to the techniques performed, exhausting the subject without returning to the previous topic.
5- In relation to the tables that indicate Specificity, Sensitivity and Accuracy of each technique separately, it would be better to generate just one with all this data indicating the results observed by VITEK2, microarray and sequencing for each class of antimicrobial. This would make it easier to view comparative data.
6- In line 195, sample EG-98203 is cited. In line 228, the same sample is called EG-98213.
7- In line 249 it is said that discrepancies are observed. What are these? It's not clear.
8- Check whether all genes have their partial spelling in italics. For example, in line 428 this is not observed.
9- Finally, I suggest revising the text not only to avoid repetitive expressions but also to improve your English.
Author Response
Comment 1:
“The title suggests that the work aims to establish an Epidemiological Platform for identifying Carbapenem Resistance in Egypt using Microreact and Microarray for Real-Time Surveillance. But although the text emphasizes carbapenems, other classes are discussed and the importance of patient data, such as age, is evaluated. In fact, the work indicates, in a very relevant way, the epidemiological analysis of E. coli infection in Alexandria, Egypt, between 2020 and 2023 using phenotypic and genotypic tools. I suggest adapting the title.”
Response to Comment 1:
We thank the reviewer for the valuable suggestion regarding the title. We have revised the title to better reflect the scope of our work, including the emphasis on multidrug resistance across various antibiotic classes, the integration of patient demographic data, and the epidemiological analysis conducted. The new title is: “Tracking Multidrug Resistance in Gram-Negative Bacteria in Alexandria, Egypt (2020–2023): An Integrated Analysis of Patient Data and Diagnostic Tools.”
Comment 2:
“A graph or table would be clearer to indicate the O/H distribution of E. coli.”
Response to Comment 2:
We appreciate the reviewer’s suggestion to enhance the clarity of the O:H distribution of E. coli. In response, we have expanded Figure 1 to include detailed panels: Figure 1a now illustrates the O:H distribution, Figure 1b presents MLST data, and Figure 1c shows carbapenemase gene distribution. Additionally, the text has been updated to reference these figures and include the relevant table citations for improved coherence.
Comment 3:
“Figure 1b is cited in the text as Figure 2 (Line 161).”
Response to Comment 3:
We thank the reviewer for identifying this discrepancy. The citation in Line 161 (line 178 in the revised version) has been reviewed, and we confirm that Figure 2 is correct in this context, as it appropriately presents all strains and their carbapenemase genes side-by-side. The text has been carefully checked to ensure consistent and accurate figure citations throughout the manuscript.
Comment 4:
“To make the text easier, I suggest listing the cases of agreement and then the exceptions for each class of antimicrobials in relation to the techniques performed, exhausting the subject without returning to the previous topic.”
Response to Comment 4:
We thank the reviewer for the suggestion to improve the clarity of the Results section. In response, we have restructured this section for better readability and logical flow. The revised structure begins with O:H typing, followed by MLST data, which has been combined with previously redundant parts for conciseness. Next, we present the detected resistance genes, transitioning into a detailed comparison of Array vs. Sequencing and Genotype vs. Phenotype results, exhaustively addressing agreements and exceptions without revisiting earlier topics. Finally, we provide an expanded discussion of the Phylogenetic Tree, highlighting its significance. We believe these changes enhance the coherence and comprehensibility of the Results section.
Comment 5:
“In relation to the tables that indicate Specificity, Sensitivity and Accuracy of each technique separately, it would be better to generate just one with all this data indicating the results observed by VITEK2, microarray and sequencing for each class of antimicrobial. This would make it easier to view comparative data.”
Response to Comment 5:
We thank the reviewer for the valuable suggestion to consolidate the tables indicating specificity, sensitivity, and accuracy into a single table for improved clarity. In response, we have created a comprehensive table summarizing the comparative data of microarray versus sequencing results and genotype versus phenotype results. This unified format allows for easier visualization and comparison of the results. We hope this adjustment meets the reviewer’s expectations and enhances the manuscript's clarity.
Comment 6:
“In line 195, sample EG-98203 is cited. In line 228, the same sample is called EG-98213.”
Response to Comment 6:
We thank the reviewer for pointing out the mention of sample EG-98203 in Line 195 (in the revised version line 223) and EG-98213 in Line 228 (in the revised version line 245). We confirm that these are two different samples, both of which are examples of low accuracy observed in the comparison between microarray and sequencing. The text has been reviewed and clarified to ensure this distinction is clear and consistent throughout the manuscript.
Comment 7:
“In line 249 it is said that discrepancies are observed. What are these? It's not clear.”
Response to Comment 7:
We thank the reviewer for highlighting the lack of clarity regarding the discrepancies mentioned in Line 249 (in the revised version line 263). Upon review, we found that only one discrepancy was observed in sample EG-98192. To address this, we have removed the statement about discrepancies to align with the reviewer’s feedback and ensure the text is accurate and concise.
Comment 8:
“Check whether all genes have their partial spelling in italics. For example, in line 428 this is not observed.”
Response to Comment 8:
We thank the reviewer for pointing out the inconsistency in gene formatting. We have carefully reviewed the manuscript to ensure that all gene names, including bla in all beta-lactamases, are correctly italicized throughout the text. This has been thoroughly checked and corrected where necessary.
Comment 9:
“Finally, I suggest revising the text not only to avoid repetitive expressions but also to improve your English.”
Response to Comment 9:
We thank the reviewer for the suggestion to revise the text for improved language and clarity. The manuscript has been thoroughly reviewed by a native English speaker, and improvements have been implemented where necessary to enhance readability and avoid repetitive expressions. We hope the revised version meets the required standard.
Reviewer 2 Report
Comments and Suggestions for Authors
The manuscript authored by Braun et al has developed a combined epidemiological surveillance tool utilizing Microreact software and molecular microarray technology to track and analyze carbapenem-resistant E. coli strains in Egypt. The results demonstrated the application of multidisciplinary technologies in monitoring antibiotic resistance. This integrated approach not only improves the accuracy of data but also provides feasible solutions for resource-limited environments. The study focused on the hospital environment in Egypt, filling the gap in the epidemiology of multidrug-resistant bacteria in North Africa. By analyzing in detail the prevalence of carbapenem-resistant E. coli in Egyptian hospitals, valuable regional data was provided. I would recommend a minor revision before accepting this manuscript.
Major comments:
(1) My major concern of this study is the limited sample size. The study only analyzed 43 samples of E. coli strains, which is a relatively small sample size, and may affect the generalizability and representativeness of the results. Larger scale studies may be better able to verify the reliability of these findings. Please justify the usage of only 43 samples to develop and validate the performance of the combined surveillance methods.
(2) The study was limited to two hospitals in one city in Egypt, lacking extensive coverage of other regions. This may restrict the external validity and generalizability of the study results.
(3) Please discuss the application feasibility of your developed approaches in monitoring antimicrobial resistance to other drugs.
Minor comments:
(1) The font in Table 1 is not consistent.
Author Response
Comment 1:
“My major concern of this study is the limited sample size. The study only analyzed 43 samples of E. coli strains, which is a relatively small sample size, and may affect the generalizability and representativeness of the results. Larger scale studies may be better able to verify the reliability of these findings. Please justify the usage of only 43 samples to develop and validate the performance of the combined surveillance methods.”
Response to Comment 1:
We appreciate the reviewer’s concern regarding the sample size of this study. To address this:
- We have emphasized that the platform's foundational role begins with 43 E. coli strains, which were specifically selected to validate the microarray method by benchmarking it against sequencing results.
- The Introduction (lines: 123-141), Discussion (lines: 590–608), and Conclusion (lines: 723-754) sections have been revised to highlight the scalability of this platform as a collaborative tool, especially for resource-limited settings where comprehensive datasets may not be initially available.
- To address the issue of generalizability, we have included a vision for future cooperation, aiming to expand the platform with additional data from ongoing collaborations, enhancing its reliability and scope.
- Finally, the manuscript discusses how microarray technology offers an accessible alternative to sequencing in countries with limited resources, demonstrating its importance for addressing multidrug resistance in Gram-negative bacteria.
We believe these revisions adequately justify the use of the initial sample set while positioning the study as a scalable and impactful initiative.
Comment 2:
“The study was limited to two hospitals in one city in Egypt, lacking extensive coverage of other regions. This may restrict the external validity and generalizability of the study results.”
Response to Comment 2:
We appreciate the reviewer’s observation regarding the geographical scope of the study. While this study focuses on two hospitals in Alexandria, Egypt, the aim was to establish a foundational platform for multidrug resistance surveillance. To address this limitation, we have revised the manuscript to highlight the following points:
- The end of the Discussion section emphasizes that this study serves as a proof of concept, demonstrating the utility of microarray technology in conjunction with sequencing for resistance tracking in a localized setting (line 604).
- We have included a vision for expanding the platform to incorporate data from additional regions within Egypt and neighboring countries, thereby increasing its generalizability and external validity.
- References to past collaborations with colleagues in Greece, Romania, and Egypt further underline the potential for regional integration and scalability of the platform for broader epidemiological surveillance (lines: 131-133).
- The manuscript discusses how the inclusion of data from resource-limited settings can enhance understanding of resistance patterns globally, particularly in areas where such surveillance tools are not yet established.
We hope these revisions clarify the study’s initial focus while illustrating its potential for broader application in future collaborations.
Comment 3:
“Please discuss the application feasibility of your developed approaches in monitoring antimicrobial resistance to other drugs.”
Response to Comment 3:
We appreciate the reviewer’s suggestion and have incorporated a discussion addressing the application feasibility of our developed approaches in monitoring antimicrobial resistance (AMR) to other drugs. The DNA microarray-based detection system, as demonstrated in our study, provides a robust, rapid, and cost-effective platform for identifying AMR genes beyond carbapenem resistance. Its adaptability lies in its design flexibility, allowing the inclusion of probes targeting resistance genes associated with other classes of antibiotics, such as beta-lactams, aminoglycosides, fluoroquinolones, and polymyxins.
We have updated the end of discussion (line 590-597) to reflect these points and included a discussion in the conclusion section, highlighting the feasibility and potential applications of our approach in monitoring AMR to a wide range of antimicrobial drugs.
Minor comments 1
“The font in Table 1 is not consistent.”
Response to minor Comment 1:
We thank the reviewer for pointing out the inconsistency in the font used in Table 1. The font has been reviewed and corrected to ensure consistency throughout the table. This adjustment enhances the table's overall presentation and aligns it with the formatting standards of the manuscript.